

# Intestinal parasitic infection alters bacterial gut microbiota in children

Miguel A. Toro-Londono[1], Katherine Bedoya-Urrego[1,2], Gisela M. Garcia-Montoya[2], Ana L. Galvan-Diaz[3] and Juan F. Alzate[1,2]

[1] Centro Nacional de Secuenciación Genómica—CNSG, Universidad de Antioquia, Medellin, Antioquia, Colombia

[2] Parasitology group, School of Medicine, Universidad de Antioquia, Medellin, Antioquia, Colombia

[3] Environmental Microbiology Group, School of Microbiology, Universidad de Antioquia, Medellin, Antioquia, Colombia

## ABSTRACT

The study of the burden that parasites can exert upon the bacterial gut microbiota was restricted by the available technologies and their costs. Currently, next-generation sequencing coupled with traditional methodologies allows the study of eukaryotic parasites (protozoa and helminths) and its effects on the human bacterial gut microbiota diversity. This diversity can be altered by a variety of factors such as age, diet, genetics and parasitic infections among others. The disturbances of the gut microbiota have been associated with a variety of illnesses. Children population in developing countries, are especially susceptible to parasitic infections because of the lack of proper sanitation and undernutrition, allowing both, the thriving of intestinal parasites and profound alteration of the gut microbiota. In this work, we have sampled the stool of 23 children from four different children's care-centers in Medellin, Colombia, and we have identified the eukaryotic parasites by traditional and molecular methodologies coupled with microbial profiling using 16S rDNA sequencing. This mixed methodology approach has allowed us to establish an interesting relationship between *Giardia intestinalis* and helminth infection, having both effects upon the bacterial gut microbiota enterotypes, causing a switch from a type I to a type II enterotype upon infection.

## INTRODUCTION

The community of hundreds of bacteria species (trillions of individual bacteria) inhabiting the human gastrointestinal tract, plays a paramount role in human health. These microorganisms referred collectively as the gut microbiota (GM) have evolved in association with themselves and with us for millions of years. Human hosts depend upon GM for (i) the acquisition of key nutrients from food, (ii) the shaping of the immune system and (iii) protection from opportunistic pathogens (*Greenblum, Turnbaugh & Borenstein, 2012*; *Huttenhower et al., 2012*; *Petrof et al., 2013*).

It is estimated that there are –at least–400 different species of bacteria in the human gastrointestinal tract (*Eckburg et al., 2005*; *Huttenhower et al., 2012*). Despite this high

Corresponding author
Juan F. Alzate,
jfernando.alzate@udea.edu.co

number of distinct bacterial taxa, they belong to a relatively low number of phyla (*Ley, Peterson & Gordon, 2006*; *Ley et al., 2008*). To date, more than 50 bacterial phyla have been identified, among them, Bacteroidetes, Firmicutes, Actinobacteria, and Proteobacteria are the most abundant in the human gut. The relative abundance of each one of them might vary between individuals and populations (*Blaser & Falkow, 2009*; *Huttenhower et al., 2012*; *Bär et al., 2015*).

In recent years, high-throughput technologies—like next-generation sequencing—have allowed an in-depth view of the composition of the GM, its genes, functions, metabolites, and proteins (*Morgan & Huttenhower, 2014*). These new technologies and the use of animal models, has shown the importance of the GM when it comes to conferring protection against pathogens and how its dysbiosis (see below), such as those caused by the use of antibiotics, leads to disease (*Ubeda & Pamer, 2012*; *Ubeda, Djukovic & Isaac, 2017*).

In a healthy state, the GM behaves like a microbial system in which the commensal (or mutual) organisms and the gut mucosal immune system co-exist, evoking the second minimal to non-inflammatory response upon the first. Several factors as age, genetics, xenobiotics, infections and even diet, has a profound impact in the gut microbiota ecosystem (*Lynch & Pedersen, 2016*; *Syal, Kashani & Shih, 2018*). It has been suggested that the human bacterial gut microbiota (BGM) composition can be divided into three categories or enterotypes (*Arumugam et al., 2011*). These enterotypes are usually referred to by the most abundant organism present in a given individual: *Bacteroides* spp. (Enteroype I), *Prevotella* spp. (Enterotype II) and members of the order Clostridiales (Enterotype III) (*Arumugam et al., 2011*; *Wu et al., 2011*). The fact that inflammatory bowel disease (IBD) do not develop in germ free experimental animals (*Sellon et al., 1998*), diverted the interest of finding a particular pathogen causing IBD and instead, it shifted over the concept of "dysbiosis" or the imbalance between the beneficial and harmful bacteria in the GM as the main IBD cause (*Darfeuille-Michaud et al., 1998*; *Autschbach et al., 2005*; *Selby et al., 2007*; *Syal, Kashani & Shih, 2018*). Various metagenomic analysis has shown in fact, a reduction in the diversity of the BGM in patients with IBD (*Manichanh et al., 2006*). Besides IBD, extra-intestinal autoimmune disorders like multiple sclerosis and rheumatoid arthritis have also been linked with intestinal dysbiosis, suggesting that BGM may play a role in the development and progression of inflammatory diseases (*Belkaid & Hand, 2014*). Other conditions that can be associated with the alterations of GM are obesity, type 2 diabetes, and autism (*Marzano et al., 2017*).

The human GM is a complex ecosystem encompassing besides bacteria; viruses (mostly bacteriophages), fungi, helminths and protozoa which interact and compete with each other (*Filyk & Osborne, 2016*). In order to understand the crosstalk between the different members of the microbiota such as bacteria and eukaryotes, and its effects on the host health/disease processes, it is increasingly important to adopt approaches that allow the overall study of the communities present in human intestines (*Chabé, Lokmer & Ségurel, 2017*).

The effects that parasitic protozoa (*Blastocystis* spp., *Giardia intestinalis*, *Entamoeba* spp., *Cryptosporidium* spp., etc.) and metazoa (roundworms, whipworms, pinworms, threadworms, hookworms and tapeworms) have on human BGM, has been left-out in

most of the studies because of the particular characteristics of industrialized countries, in which development—it is thought—has depleted the number of parasite-infected individuals and the diversity of the gut microbiota mostly by diet, improved sanitation, the use of food sterilization and antibiotics (*Martínez et al., 2015*; *Chabé, Lokmer & Ségurel, 2017*). It is known that the load of these parasites in the population of industrialized countries affects the diversity of the BGM when compared with the few studies carried in non-industrialized counterparts (*Chabé, Lokmer & Ségurel, 2017*). Whether the impact of these changes—increased BGM diversity in non-industrialized countries versus lower diversity in industrialized ones—are beneficial for the overall health status of an individual or not, is still to be determined (*Blaser & Falkow, 2009*; *Segata, 2015*; *Chabé, Lokmer & Ségurel, 2017*). It is also noteworthy that these studies take into account either the protozoa or the metazoa, but not both at the same time.

When it comes to helminths, for example, it is estimated that more than 1 billion people worldwide are infected by soil-transmitted helminths such as *Ascaris lumbricoides* and *Trichuris trichiura* among others. In humans, a study in Sri Lanka in 2017 showed that there is indeed an association with GM diversity in people infected with helminths (*Jenkins et al., 2017*). It is well known that helminthic parasites release various excretory-secretory products such as immunomodulatory proteins, glycoproteins, and miRNAs that modulate the activity of different cell types including regulatory immune cells. The effects that helminths have on the immune system are so drastic that they have been used for therapeutic purposes (*Sipahi & Baptista, 2017*). In developing countries, the children population is especially susceptible to parasite infections due to socioeconomic conditions—like poor sanitation–that allow the transmission of gastrointestinal parasites. Besides, it is known that malnutrition has a synergistic relationship with gastrointestinal infections related to the loss of integrity of the gastrointestinal mucosa (*Peterson & Artis, 2014*). Interestingly enough, malnutrition is a risk factor and a consequence of intestinal protozoal infection (*Ibrahim et al., 2017*).

Just like helminths, protozoa have been blatantly neglected. Along with rotavirus and bacteria, protozoa parasites are one of the leading causes of diarrhea. In 2010, approximately 357 million cases of diarrhea were caused either by *Entamoeba histolytica.*, *Cryptosporidium* spp.*, or *Giardia intestinalis* (*Naghavi et al., 2015*; *Gilchrist et al., 2016*; *Burgess et al., 2017*). In the case of *Cryptosporidium* spp., it has been established as the second aetiological agent of diarrhea in children under 11 months in sub-Saharan Africa and south Asia, and the third for children between 12 and 23 months (*Kotloff et al., 2012*). Additionally, in the MAL-ED study carried out in South America, Africa, and Asia, researchers have a found *Cryptosporidium* spp. as the fifth most frequent pathogen in stool samples of children in the ranges of 0–11 and 12–24 months (*Platts-Mills et al., 2015*). It has been revealed that at least 15 different protozoa genera from diverse groups, either parasitize or commensalize the human gut (*Hamad, Raoult & Bittar, 2016*).

It has been hypothesized that in humans the infection by *G. intestinalis* triggers long-lasting alterations of the commensal microorganisms, promoting bacterial invasiveness in the gut mucosa during its post-clearance phase. This was revealed by a study carried out in mice that showed that damage in the epithelial barrier, cause a host unresolved

immune reaction towards its microbiota (*Chen et al., 2013*). A study that took into account *Blastocystis* spp.*, Entamoeba* spp.*,* and *G. intestinalis* in human population in Côte d'Ivoire in 2016, showed a grouping separation between the individuals infected with *Giardia* and the other parasites, indicating that the microbial communities could be reshaped solely by the presence of *Giardia* parasites (*Iebba et al., 2016*). These studies suggest that like helminths, protozoa parasites affect the composition of the human bacterial gut microbiota.

In Colombia, the latest survey of parasitic diseases in school-aged children revealed that 30% of the subjects have soil-transmitted helminths, being the most abundant *T. trichiura* (18%) and *A. lumbricoides* (11%). Regarding protozoa, *Blastocystis* spp. (which is no longer classified as protozoa) is the most abundant (52%) followed by *Entamoeba* spp. (17%)*, G. intestinalis* (15%) and *Cryptosporidium* spp. (0.5%). Results of our own survey in Medellín (Colombia) (about to be published), estimates a higher prevalence of *G. intestinalis* (26%) and *Cryptosporidium* spp. (1.4%) compared with the most recent national survey (*Ministerio de Salud y Protección Social and Universidad de Antioquia, 2014*).

This work is a study of the gut microbiota disturbances, considering protozoa and helminths of 23 children with parasitic infections from Medellín, Colombia. Both the protozoa and metazoa parasites were identified by microscopy from the stool samples and in the case of *Cryptosporidium* spp., the species were identified using molecular techniques. The microbiota from the stool was examined by high-throughput sequencing of the V4 variable region of the bacterial 16S rDNA gene followed by bioinformatics and statistical analysis of the sequenced data. To our knowledge, this is one of the few studies that encompasses bacteria, protozoa, and metazoa of the human gut microbiota.

## MATERIALS AND METHODS
### Study area and sample population
A cross sectional study with non-probabilistic sampling was carried out during 2015 in seven different daycare centers in Medellín, Colombia. All day-care centers are government funded and the children in the study receive a well-balanced diet. Two hundred and ninety feces samples were collected. After parasitical analysis with conventional microscopic techniques, 23 fecal samples were selected for the microbiota analysis. The selection criteria included that only one parasite was detected in order to reduce noise introduced by cross-infections. In the case of helminth infection, it was not possible to have samples with one single helminth. Therefore, we decided to circumvent this situation by selecting a group of children infected with *Ascaris lumbricoides* but they were also infected with either *Trichuris trichiura* or *Enterobius vermicularis* and most of them were also positive to non-parasitic protozoa or *Giardia*. This selection criteria reduced the number of groups and samples as follows: (i) Group Crypto (*Cryptosporidium* detected, no other parasites observed): five individuals; (ii) Group Giardia (*Giardia* detected, no other parasites observed): six children; (iii) Group Helm-pro (*Ascaris detected,* and were coinfected with either *Trichuris trichiura* or *Enterobius vermicularis*. Most of them with accompanying non-parasitic protozoa or *Giardia*) six children; and (iv) Group control (No parasites detected): six children (Table 1).

**Table 1 Description of samples used in this study.** Different features like origin of the sample, age and sex, along with the different parasites found in each sample, are indicated.

| Care-centers | Group | Sample | Sex | Age | Parasites observed by microscopy or PCR |
|---|---|---|---|---|---|
| | Helm-Pro | CD13 | F | 5 | *Entamoeba h/d/m; Blastocystis* spp; *Ascaris lumbricoides; Trichuris trichiura* |
| Care-center 1 | Crypto | CD17 | F | 2 | *Cryptosporidium; Blastocystis* spp |
| | Control | CD18 | M | 3 | *none* |
| | Giardia | ED05 | M | 2 | *Giardia intestinalis* |
| | Crypto | ED06 | F | 1 | *Cryptosporidium* |
| | Crypto | ED15 | M | 1 | *Cryptosporidium; Blastocystis* spp |
| | Giardia | ED16 | F | 1 | *Giardia intestinalis* |
| | Helm-Pro | ED17 | F | 2 | *Blastocystis spp; Ascaris lumbricoides; Enterobius vermicularis* |
| | Helm-Pro | ED23 | M | 3 | *Giardia intestinales; Chilomastix mesnilii; Ascaris lumbricoides; Trichuris trichiura* |
| | Control | ED29 | F | 2 | *none* |
| Care-center 2 | Crypto | ED33 | F | 1 | *Cryptosporidium* |
| | Control | ED36 | F | 4 | *none* |
| | Giardia | ED47 | F | 3 | *Giardia intestinalis* |
| | Giardia | ED54 | F | 4 | *Giardia intestinalis* |
| | Helm-Pro | ED69 | F | 5 | *Entamoeba hartmanni; Giardia intestinalis; Blastocystis* spp; *Ascaris lumbricoides* |
| | Giardia | ED77 | M | 3 | *Giardia intestinalis* |
| | Crypto | ED9 | M | 1 | *Cryptosporidium* |
| | Control | ES02 | M | 5 | *none* |
| Care-center 3 | Giardia | ES04 | N.A | N.A | *Giardia intestinalis* |
| | Control | ES05 | M | 3 | *none* |
| | Control | ES07 | M | 5 | *none* |
| Care-center 4 | Helm-Pro | SE23 | F | 4 | *Blastocystis* spp.; *Ascaris lumbricoides; Trichuris trichiura* |
| | Helm-Pro | SE56 | M | 4 | *Giardia intestinalis; Ascaris lumbricoides; Trichuris trichiura* |

## Fecal sample collection, parasitical analysis, DNA extraction

Stool samples were collected in screw-capped containers (no preservative) and immediately transported to the lab where they were analyzed by two methods, direct microscopic examination and modified Ritchie concentration. The modified Ziehl-Neelsen stain was used for detection of intestinal apicomplexan. The samples were frozen at −20 °C prior DNA extraction for microbiota analysis and *Cryptosporidium* spp. assignation. *Cryptosporidium* was detected by a nested PCR of an approximately 830-bp fragment of the small-subunit (SSU) rRNA gene, according to the protocol described by *Xiao et al. (1999)* and *Xiao et al. (2001)*, The DNA was purified using a stool DNA isolation kit (Norgen Biotek, Thorold, ON, Canada), according to manufacturer's instructions.

## Amplification and sequencing of 16S rDNA variable 4 region

DNA concentration was assessed using fluorescent dye picogreen. Additionally, the quality of bacterial DNA was tested using an initial amplification of the full-length 16S rDNA gene, using 27F and 1492R primers. The samples were sequenced by the Illumina MiSeq platform

with paired reads of 250 bp. where the V4 hypervariable region of the 16S rDNA gene was amplified using the Hyb515F_rRNA (GTTTGATCMTGGCTCAG) and Hyb806_rRNA (TGCCTCCCGTAGGAGT) primers with the respective Illumina adapters and barcodes.

Raw sequencing data was deposited at the NCBI SRA database under the project PRJNA487588.

## Sequenced 16S rDNA processing and analysis

MOTHUR software package (v 1.39.5) was used for sequenced data processing according to the MiSeq Standard Operating Procedure (SOP). Low quality and nonspecific reads were detected and excluded with the screen.seqs command. The chimeric sequences were identified and discarded using the VSEARCH program included in the package. The Operations Taxonomic Units (OTUs) were defined at 97% followed by taxonomy assignment using the RDP reference database sequences as a guide. Singletons and OTUs, identified as both chloroplasts and mitochondria, were discarded.

## Alpha & Beta diversity metrics

Sample coverage (good's coverage), richness (CHAO index), diversity (Shannon and Inverse Simpson index) and the number of observed OTUs, were obtained with MOTHUR (Table 2). To visualize differences in microbial community structures based on species abundances, $\theta_{YC}$ dissimilarity matrices were generated from OTU tables and subsequently subjected to PCoA analysis and visualized in R. $\theta_{YC}$ dissimilarities were calculated using MOTHUR and used to compare samples, upon which analysis of molecular variance (AMOVA) were performed using the MOTHUR integrated version (*Schloss et al., 2009*). The OTU tables from MOTHUR were exported as a BIOM file and used in the Metagenomics Core Microbiome Exploration Tool (MetaCoMET) to generate the diversity plots (*Wang et al., 2016*). The images were further modified using Inkscape.

## Ethics committee approval

The ethical clearance of this study was followed by the ethics of Helsinki declaration and resolution No. 008430 of 1993 from the Ministry of Health from Colombia. The study was approved by the Ethics Committee from Sede de Investigación Universitaria, Universidad de Antioquia, under the official document No. 14-06-564.

## RESULTS

From the 290 collected fecal samples, nearly 39% were negative for intestinal parasites and 23 were selected for this study according to the parasites detected. The samples were grouped as follow: (i) Controls: the samples in which it was not possible to detect any known parasite; (ii) Crypto: the samples of children that were infected by *Cryptosporidium* spp. Some of them also had non-parasitic protozoa; (iii) Giardia: the samples of children infected with *Giardia intestinalis* and (iv) Helm-Pro: mixed samples of children infected by *Ascaris lumbricoides*, some of them with either *Trichuris trichiura* or *Enterobius vermicularis* and most of them with accompanying non-parasitic protozoa or *Giardia* sp. All the samples with helminthic parasites were accompanied by either protozoa or blastocystis, the reason

**Table 2  Statistical diversity of the individuals surveyed in the study.**

| Sample | Number of sequences | Coverage | OTUs | Simpson index | Shannon index | CHAO index |
|---|---|---|---|---|---|---|
| ES02 | 3,319 | 0.983 | 162 | 16.214 | 3.554 | 228.500 |
| ED36 | 9,125 | 0.992 | 199 | 9.971 | 3.060 | 325.136 |
| CD18 | 6,949 | 0.992 | 152 | 15.401 | 3.247 | 251.750 |
| ES04 | 5,652 | 0.992 | 126 | 4.231 | 2.320 | 180.474 |
| ED54 | 7,722 | 0.992 | 171 | 7.270 | 2.968 | 318.500 |
| ED47 | 6,339 | 0.993 | 160 | 15.064 | 3.508 | 217.400 |
| ED69 | 17,814 | 0.994 | 343 | 5.511 | 3.081 | 490.163 |
| ED17 | 13,114 | 0.994 | 220 | 12.430 | 3.340 | 382.048 |
| ED16 | 8,427 | 0.994 | 164 | 5.556 | 2.936 | 246.875 |
| CD13 | 1,8735 | 0.995 | 229 | 2.367 | 1.950 | 419.120 |
| ED06 | 15,928 | 0.995 | 221 | 22.718 | 3.599 | 395.789 |
| ED23 | 20,283 | 0.995 | 257 | 9.080 | 3.095 | 512.316 |
| SE23 | 19,904 | 0.995 | 262 | 4.362 | 2.455 | 400.182 |
| ES07 | 13,312 | 0.995 | 224 | 16.021 | 3.435 | 324.800 |
| ED05 | 5,472 | 0.995 | 95 | 6.351 | 2.518 | 124.545 |
| ED15 | 18,153 | 0.996 | 177 | 4.207 | 2.495 | 420.077 |
| ES05 | 1,8545 | 0.996 | 215 | 10.020 | 3.124 | 335.120 |
| ED77 | 17,978 | 0.996 | 212 | 6.857 | 3.002 | 336.250 |
| CD17 | 22,208 | 0.996 | 201 | 3.580 | 2.270 | 329.375 |
| SE56 | 57,133 | 0.997 | 360 | 4.276 | 2.276 | 901.750 |
| ED9 | 20,140 | 0.997 | 153 | 15.584 | 3.326 | 354.600 |
| ED29 | 20,649 | 0.997 | 150 | 9.368 | 2.713 | 294.000 |
| ED33 | 10,385 | 0.998 | 70 | 6.552 | 2.436 | 100.000 |

why it is not possible to have a group of individuals infected solely by helminths. The groupings and other features of each sample are indicated in Table 1.

After the bioinformatic processing of the reads with the MOTHUR package, the number of clean sequences (excluding low-quality, too short, non-bacterial and chimeric reads) in the studied samples ranged between 3,319 and 57,133. The Good's coverage estimator values of the samples ranged from 0.982 to 0.997.

The Shannon index—which shows the diversity of species in each community—present significative difference between the assessed groups (Fig. 1A). Though this result seems counter-intuitive because the Shannon indexes changes observed in parasite infected individuals, this low Shannon among all the groups might be associated with the age of the population (<5 y/o). Interestingly enough, the number of observed species in the Helm-pro group is higher than those found in the other groups, suggesting that the infection with helminths (more specifically nematodes) have an effect on the total number of bacterial species observed in the children gut (Fig. 1B).

When the BGM is compared between groups at genera level (Fig. 2), there is a evident change between the Control and the Helm-Pro and Giardia groups with an enrichment of *Prevotella* spp. at expense of *Bacteroides* spp. Additionally, as Fig. 3 shows, a comparison of

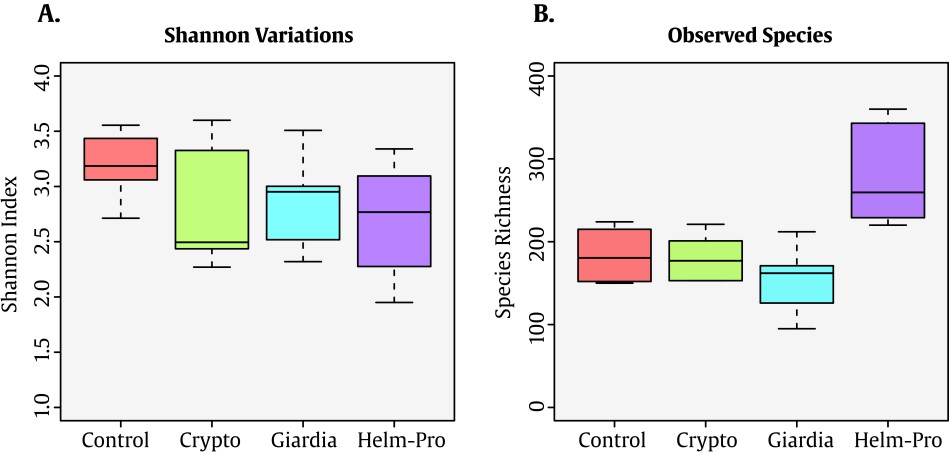

**Figure 1  Box-plot comparisons of bacterial diversity and species richness.** (A) Bacterial diversity of the different groups using Shannon's alpha index and (B) Species Richness found in the same groups. The top and bottom boundaries of each box indicate the 75th and 25th quartile values, respectively, and lines within each box represent the 50th quartile (median) values. Ends of whiskers mark the lowest and highest diversity values in each group. Though a difference in the Shannon index was expected between infected and not-infected individuals, the evenness observed can be attributable to the age of the children. Interestingly enough, the group with helminths and protozoa showed the highest number of bacterial species.

the microbiota of each of the samples within the different groups in all the ranks, there is a clear difference in the population composition with an enrichment of the Prevotellaceae family in Giardia and Helm-Pro groups at the expense of the Bacteroidaceae family, more abundant in the Control and some Crypto samples.

Likewise, *Prevotella* genus is the most enriched genera in the Helm-Pro and Giardia groups and the *Bacteroides* in the Controls and Crypto groups. Principal coordinates analysis (PCoA) showed an explicit grouping in the structure of the bacterial communities of the Giardia and Helm-Pro groups and the dissimilarities of these with those of the Control and Crypto (Fig. 4A).

These results combined, show a strong correlation between the change in the enterotype of the individuals when these are infected with the protozoan parasite *Giardia* sp. and helminths. These results are corroborated by the AMOVA analysis that shows a significant variance of the control with either giardia ($p$-value $= 0.006$) or helm-pro ($p$-value $= 0.014$) groups (Table 3). No other group comparison showed significant variance between them.

To help us better understand the differences underlying between the groups, we used a Venn diagram (Fig. 4B) that group the different OTUs found in all the groups and how they relate, especially those that intersect between Giardia and Helm-Pro. The two most abundant OTUs in the shared microbiota of Helm-Pro and Giardia are from *Prevotella* species (51%) and unclassified *Porphyromonadaceae* species from the bacteroidetes Phylum (Fig. 5A), likewise, in the unique Helm-Pro OTUs, the most abundant genera are also from the same phyla, *Alloprevotella* (40%) and *Prevotella* (16%) (Fig. 5B). In contrast, in the unique Giardia OTUs, the most abundant bacteria are Ruminococcus (17%) from the Firmicutes Phylum and in second place *Prevotella* species (12%) (Fig. 5C).

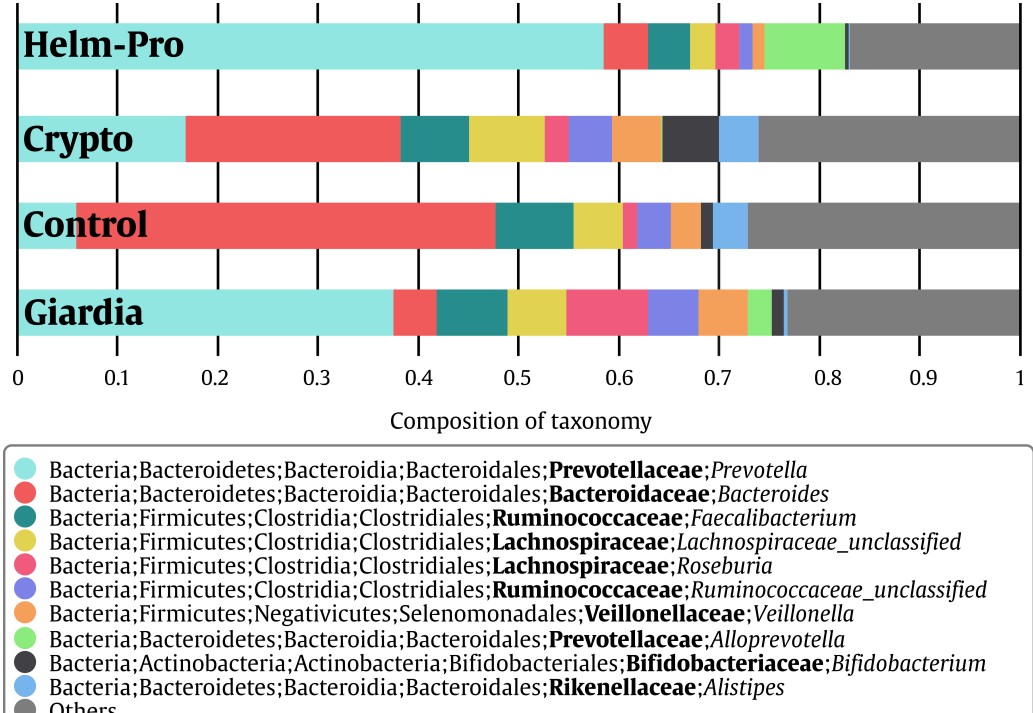

**Figure 2** **Bacterial gut microbiota comparison at Genera level per group.** A switch from enterotype I enriched in *Bacteroides* spp. in the control individuals to enterotype II enriched with *Prevotella* spp. in the Helm-Pro and Giardia groups is noticeable, suggesting a change of enterotype upon parasite infection. The families are shown in bold and the genera in italics.

## DISCUSSION

Different from the concept of pathogenicity that can be associated to a species or subspecies of bacteria, dysbiosis can be challenging to define as it could be considered a perturbation of the balanced ecology of an established system, in this case the gut microbiota.

To study the effect that parasites cause upon the bacterial community residing in the human gut, we used primers that target the 16S gene V4 hypervariable region observing up to 360 bacterial species (Fig. 1B) and Table 2).

As expected, we found the organisms that define the enterotypes to be the most common in the children studied in Medellín (Figs. 2 and 3). From our results we can conclude that the children from the control group are mainly associated with a type I enterotype (defined by *Bacteroides* spp.) which is switched to a type II enterotype (determined by *Prevotella* spp.) upon infection with helminths and *Giardia intestinalis* parasites. This is a striking result for two reasons: (i) changing enterotypes in humans is not easy to achieve (*Wu et al., 2011*; *Yin et al., 2017*), for instance, a study of 10 subjects who were subjected to a dietary change from high-fat/low-fiber to low-fat/high-fiber, did indeed modulated the BGM but they did not change the enterotype (*Wu et al., 2011*; *Roager et al., 2014*); and (ii) since the enterotypes are associated with long-term dietary habits, and these are somewhat

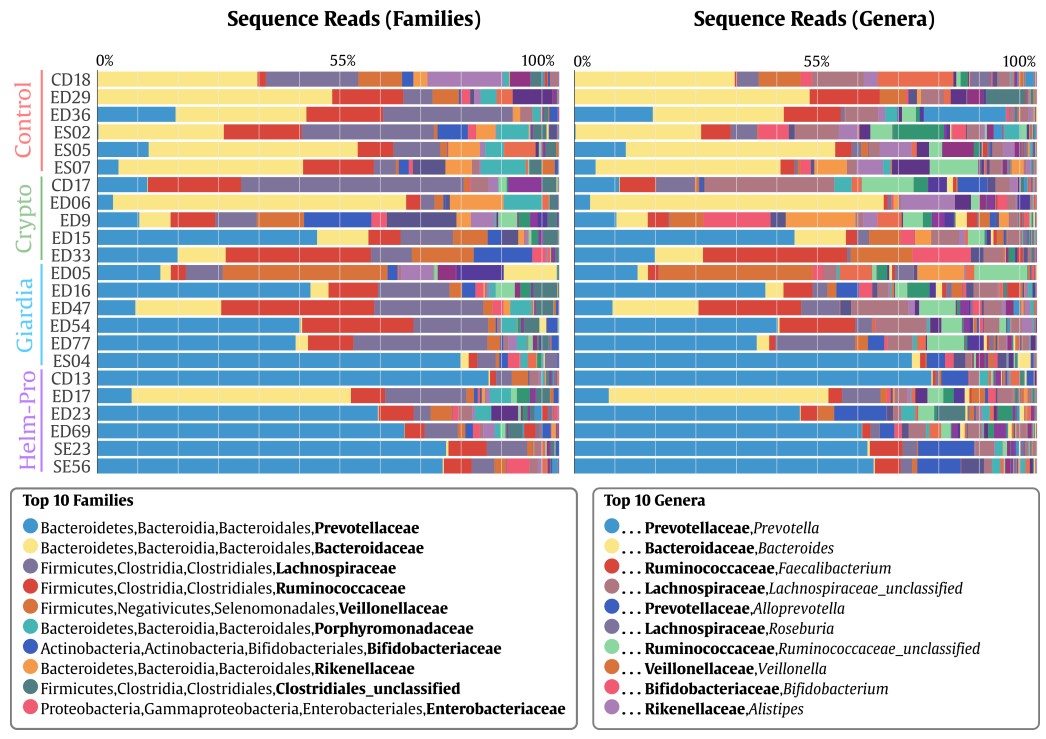

**Figure 3** **Bacterial gut microbiota comparison at Family and Genera levels per child.** Relative abundances and richness at individual resolution. An evident enrichment of the Prevotellaceae family in ED16, ED54, ED77, ES04 individuals (Giardia group) and CD13, ED23, ED69, SE23 and SE56 individuals (Helm-Pro group) is present at expense of the Bacteroidaceae family more abundant in all the individuals of the control group. The families are shown in bold and the genera in italics.

different between industrialized countries and developing ones (*De Filippo et al., 2010*), it would be reasonable to assume that populations from the same region (in this case the same city) with similar diet, would have the same enterotypes (*Yatsunenko et al., 2012*). All the sampled children were from close locations in an urban settlement within a 16 km² area. Whether the shift of enterotypes elicited by the parasite's infection is permanent or reversible, as shown in mice upon the clearance of the parasites (*Houlden et al., 2015*), is yet to be determined. We are well aware of the studies that claim that the enterotypes construction might be not that accurate, and that instead the BGM is a gradient of species (*Caporaso et al., 2011*; *Knights et al., 2014*); nevertheless, because of the simplicity of the enterotypes model compared with the gradient one and the adherence of our data to the former, we have decided to discuss our results using the enterotypes model.

The type I enterotype has been associated to individuals with a westernized diet rich in animal protein and it is the most common enterotype in industrialized countries, whereas type II enterotype is associated with developing nations and a diet rich in carbohydrates (*De Filippo et al., 2010*; *Ou et al., 2013*). In fact, *Prevotella* spp. is abundant in the BGM of cattle and goats (*Flint et al., 2008*). It is noteworthy that the human populations with

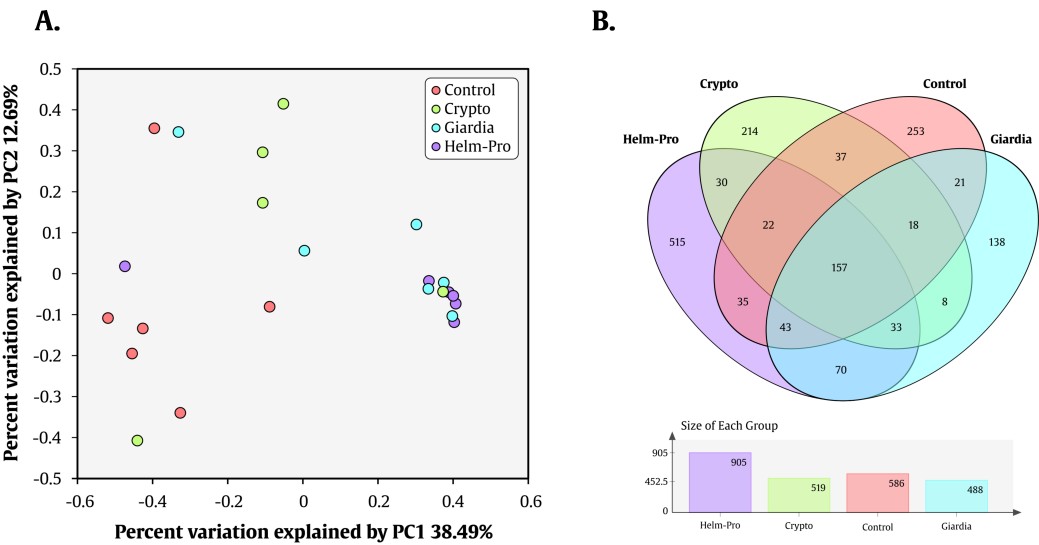

**Figure 4** **PCoA and Venn Diagram.** PCoA of the dissimilarities among bacterial communities and taxonomical structures in the control, crypto, giardia and helm-pro groups using $\theta_{YC}$ distances. There is a clustering between the individuals of the giardia group (cyan) and the helm-pro group (purple) showing that these populations are not as different as those from the control group (red). (B) Venn diagram depicting the number and OTU's distribution between the studied groups. The unique and shared OTUs of the Helm-Pro and Giardia groups could help to explain the special characteristics and structure of the parasite infected BGM.

high prevalence of helminths studied so far are very distinct in lifestyle and diet from industrialized areas (*De Filippo et al., 2010*; *Reynolds, Finlay & Maizels, 2015*).

The type I and II enterotypes behave differently in fiber degradation and metabolites production. For instance, type II enterotype is more prone to produce propionate and butyrate as by-products of fiber degradation, being the latest a known inflammatory modulator (*Chen et al., 2017*). This is particularly interesting considering that it has been suggested that a potential mechanism by which helminths alter the BGM composition is affecting the host immune system disrupting the interaction between the bacteria and the host (*Resende Co et al., 2007*; *Degarege et al., 2012*; *Brosschot & Reynolds, 2018*). For example, it is known that in rats *Hymenolepis diminuta* establishes long-term colonization that has an immunomodulatory effect on the host without causing bacterial dysbiosis (*Parfrey et al., 2017*). Another impact of helminthic infection is either a decrease or an increase of species observed in mice and humans respectively (*Lee et al., 2014*; *Houlden et al., 2015*).

Nevertheless, in our study there is not a clear distinction between the controls and the infected groups (Fig. 1A), this can be attributable to the young age of the individuals surveyed as it has been shown that the BGM diversity in children is low and increases with age (*Odamaki et al., 2016*). In humans, a study in Sri Lanka in 2017 showed that there is indeed an association with GM diversity in people infected with helminths (*Jenkins et al., 2017*). Previous studies had shown a discrepancy between the effect of *Trichuris trichiura* on the GM; on 2013 *Lee et al. (2014)* demonstrated an increase in diversity and abundance of

**Table 3 Treatment of the data by AMOVA analysis.** Only Control/Giardia and Control/Helm-Pro showed significant variations.

| Hypothesis | Source of variation | Sum of squares | Degrees of freedom | MS | Fs | *p*-value |
|---|---|---|---|---|---|---|
| Control/Crypto | Among | 0.47 | 1 | 0.47 | | |
| | Within | 2.90107 | 9 | 0.322341 | 1.45808 | 0.12 |
| | Total | 3.37107 | 10 | | | |
| Control/Giardia | Among | 1.01302 | 1 | 1.01302 | | |
| | Within | 2.50902 | 10 | 0.250902 | 4.03752 | 0.006* |
| | Total | 3.52204 | 11 | | | |
| Control/ Helm-pro | Among | 1.19073 | 1 | 1.19073 | | |
| | Within | 2.3152 | 10 | 0.23152 | 5.14312 | 0.014 |
| | Total | 3.50593 | 11 | | | |
| Crypto/Giardia | Among | 0.298183 | 1 | 0.298183 | | |
| | Within | 2.61592 | 9 | 0.290658 | 1.02589 | 0.406 |
| | Total | 2.91411 | 10 | | | |
| Crypto/ Helm-Pro | Among | 0.477709 | 1 | 0.477709 | | |
| | Within | 2.4221 | 9 | 0.269122 | 1.77507 | 0.082 |
| | Total | 2.89981 | 10 | | | |
| Giardia/Helm-Pro | Among | 0.140854 | 1 | 0.140854 | 0.693847 | 0.639 |

some Paraprevotellacae bacteria in individuals infected by *T. trichiura*. On the other hand, *Cooper et al. (2013)* showed that in Ecuadorian children *T. trichiura* infection does not have an effect on the gut microbiota and instead, it is *A. lumbricoides* parasite colonization the one that drives the GM changes. This is particularly relevant to this study because the helm-pro group was designed considering the presence of *A. lumbricoides* as the primary determinant in the group, even when some of the individuals also had a *T. trichiura* co-infection.

Even though there seems to be a consensus about the beneficial immunomodulatory role of helminths during infection, the role that protozoa might play or not—immunologically speaking—in the human host is not yet clear, and most of the studies made so far are in murine models. For instance, in mice the colonization by *Tritrichomonas musculis* leads to inflammasome activation, promoting the release of the pro-inflammatory cytokine IL-18 and preventing infection by bacteria (*Chudnovskiy et al., 2016*). Also, in mice, *Giardia's* cysteine secretory/excretory proteases could induce abnormalities in the biofilm architecture of the host microbiota allowing bacterial invasion. These dysbiotic microbial communities stimulate the activation of TRL signaling pathway 4 and the overproduction of proinflammatory cytokine IL-1$\beta$ (*Beatty et al., 2017*). The impact of *Giardia* infection in the BGM in mice has also been studied by *Barash et al. (2017)*. Their findings indicate a systemic dysbiosis of aerobic and anaerobic commensal bacteria, characterized by

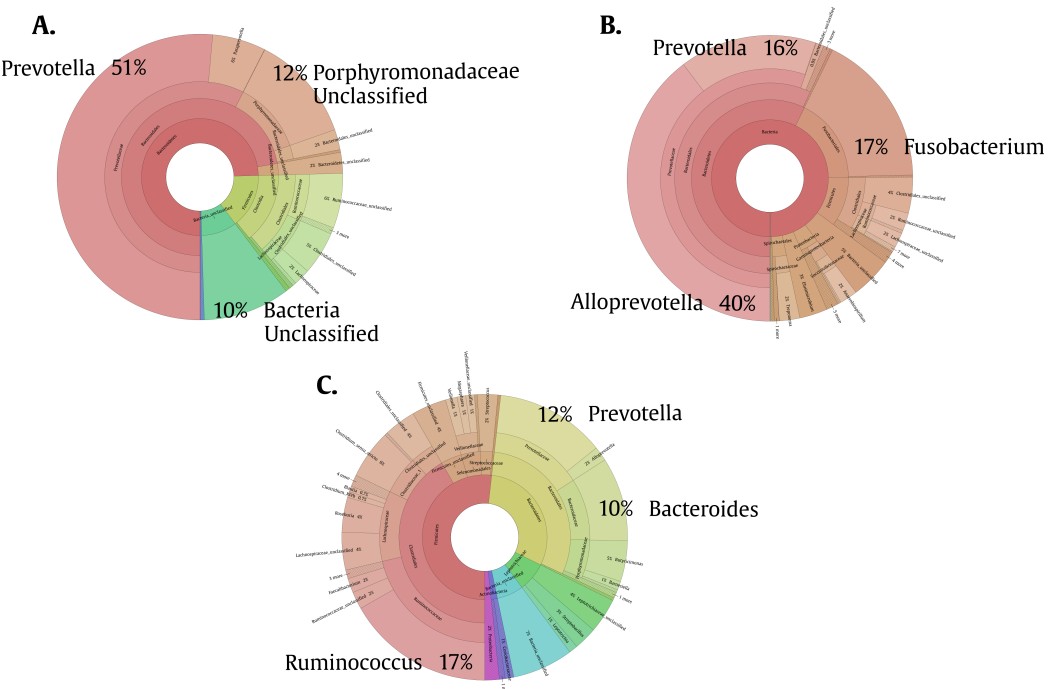

**Figure 5** **Pie charts of the three most abundant OTUs (genera) unique and shared in Giardia and Helm-Pro groups.** (A) Shared OTUs between Helm-Pro and Giardia groups. More than 50% of the genera observed in the shared OTUs belong to *Prevotella spp.*, the determinant genera of the type II enterotype. (B) Unique Helm-Pro OTUs and (C) Unique Giardia OTUs. In the unique OTUs found in both, Helm-Pro and Giardia, there is a variety of anaerobic bacteria such as *Fusobacterium* spp., and *Ruminococcus* spp.

higher levels of aerobic microorganisms and lower levels of anaerobic ones such as those from the Firmicutes phylum (*Lactobacillaceae, Eryipelotichaeae, Ruminococcus, and Clostridia*). They proposed that the dysbiosis observed could be mediated by the anaerobic *Giardia* metabolism and the stimulation of gut inflammation (*Gevers et al., 2014*; *Barash et al., 2017*).

We have found that just as observed in the Helm-pro group, the Giardia group also shifts the enterotype from I to II (Figs. 2 and 3) and, as shown by the PCoA analysis (Fig. 4A), half of the samples of this group clusters with the Helm-Pro one. Even if the mechanisms by which they exert the changes in the gut microbiota are different, the effect of the *A. lumbricoides* and *G. intestinalis* in the human BGM is somewhat similar. Opposite to the results observed by Barash in mice, we found that in the Giardia group—besides the switch to the type II enterotype—there is a higher level of *Roseburia* spp. (Fig. 2), an anaerobic bacteria known as a "maker of health" for the high butyrate production, that as mentioned before, may be necessary for the control of inflammatory processes in the gut (*Tamanai-Shacoori et al., 2017*). Another discrepancy that we found regarding the anaerobes depletion expected in Giardia infection, is that in the unique Giardia OTUs, since the most abundant (17%) is *Ruminococcus* spp., a Firmicutes phylum bacteria (Fig. 5).

If the findings of higher levels of the anaerobic *Roseburia* in the Giardia group and of Ruminococcus in the unique Giardia OTUs, are relevant or a mere curiosity of this dataset, is yet to be determined.

The last group of this study (Crypto), is formed by individuals infected with *Cryptosporidium* spp. Just as with the Giardia group, Crypto also displayed a rise in the Firmicutes phylum, especially of *Bifidobacterium* spp., which is believed to exert health benefits on their hosts (*O'Callaghan & Van Sinderen, 2016*). It has been suggested that the use of *Bifidobacterium* spp. as probiotics reduces the incidence of diarrhea in infants possibly by the competition of adherence sites on epithelial cells with other microorganisms (*Corrêa et al., 2005*; *Gueimonde et al., 2007*, *O'Callaghan & Van Sinderen, 2016*).

Nevertheless, the AMOVA analysis did not support significant differences between the Crypto and Control groups, maybe because of the microbial populations disparity displayed by the members of the former. The parasite *Cryptosporidium* spp. is rather different from helminths and *G. intestinalis* because it multiplies inside of intestinal enterocytes, so the effects that it might have upon GM could be indirect and related to the loss of integrity of the epithelium (*Moore et al., 1995*). Still, the effects that the short-lived extracellular stages of the parasite (Sporozoites, merozoites, microgametes and oocysts) can have upon the BGM cannot be wholly ruled out. In developing countries, children cryptosporidiosis also contributes to malnutrition, immunosuppression and stunting (*Guerrant, 1997*). Although studies of the effect of *Cryptosporidium* spp. on human BGM are rather lacking, encouraging rodent models help us to devise that there is indeed an effect of these parasites on the BGM (*Ras et al., 2015*). The different *Cryptosporidium* species that we found in our study group differentially in the PCoA analysis and display a different array of species abundance (Fig. 4A). For instance, there is a discrete grouping of samples of the Helm-Pro and Giardia groups with one outsider sample belonging to the Crypto group in it (Fig. 4A green circles). When it comes to the abundance profile shown in the bacterial microbiota, there is also a difference of the sample above with its Crypto group and a closer resemblance with the abundance distribution found in the Helm-Pro and Giardia groups (Fig. 4A). This particular sample (ED15) belongs to one child infected with a zoonotic avian species of *Cryptosporidium (C. meleagridis)*, the other children in the same group were infected with *C. hominis* (Table 1).

Overall, we found that the children infected with either the *Giardia intestinalis* protozoa or a mix of helminths such as *Ascaris lumbricoides* along with *Trichuris trichiura,* and other parasitic protozoa causes a switch of enterotype from I to II in children. To our knowledge, this is the first study of the effects of human parasites in human gut microbiota in an urban setting. This study in children is particularly relevant because it has been hypothesized that the disturbance in the gut microbiota by helminths can have an effect in the cognition and behavior of this susceptible population (*Guernier et al., 2017*).

## CONCLUSIONS

This study is just a glimpse into the incidence of parasite infection in the human bacterial gut microbiota in developing countries, and to draw sturdier hypothesis about the human

parasites and the BGM interaction, a more thorough study with a more comprehensive dataset will be necessary.

Our results support the observation that the presence of intestinal parasites in children, mainly *Giardia* and Helminths as *Ascaris*, exerts an effect upon the gut microbiota affecting the equilibria of bacterial communities. Nonetheless, not all parasites had the same influence on the bacterial populations, for instance, *Cryptosporidium spp.* showed no significant alterations of the bacterial microbiota in terms of diversity and structure. Future works should aim to elucidate the causal relation of the parasite arrival into the intestine and bacterial community structure changes.

### Funding
This project was funded by Vicerrectoría de Investigación, Universidad de Antioquia, "Programa de sostenibilidad de grupos 2016–2017" grant CPT1604, and Centro Nacional de Secuenciación Genómica—CNSG, sede de investigación Universitaria—SIU, Universidad de Antioquia. The funders had no role in study design, data collection and analysis, decision to publish, or preparation of the manuscript.

### Grant Disclosures
The following grant information was disclosed by the authors:
Vicerrectoría de Investigación, Universidad de Antioquia: CPT1604.
Centro Nacional de Secuenciación Genómica—CNSG.
sede de investigación Universitaria—SIU, Universidad de Antioquia.

### Competing Interests
The authors declare there are no competing interests.

### Author Contributions
- Miguel A. Toro-Londono performed the experiments, analyzed the data, prepared figures and/or tables, authored or reviewed drafts of the paper, approved the final draft.
- Katherine Bedoya-Urrego performed the experiments, authored or reviewed drafts of the paper, approved the final draft.
- Gisela M. Garcia-Montoya conceived and designed the experiments, performed the experiments, analyzed the data, prepared figures and/or tables, authored or reviewed drafts of the paper, approved the final draft.
- Ana L. Galvan-Diaz and Juan F. Alzate conceived and designed the experiments, performed the experiments, analyzed the data, contributed reagents/materials/analysis tools, prepared figures and/or tables, authored or reviewed drafts of the paper, approved the final draft.

### Human Ethics
The following information was supplied relating to ethical approvals (i.e., approving body and any reference numbers):

We followed the ethical guidelines prescribed by the Helsinki declaration and resolution No. 008430 of 1993 from the Ministry of Health from Colombia. The study was approved by the Ethics Committee from Sede de Investigación Universitaria, Universidad de Antioquia, under the official document No. 14-06-564.

### Data Availability

Data was deposited at the NCBI Sequence Read Archive (SRA) with accession number PRJNA487588. This data can also be found at: alzate, juan (2018): intestinal microbiota 16S. figshare. Fileset. Available at https://doi.org/10.6084/m9.figshare.7001399.v1.

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
