# Peer review of "Intestinal parasitic infection alters bacterial gut microbiota in children"

_PeerJ, doi:10.7717/peerj.6200_

## Round 0.1 · original submission · Major Revisions

Dear Dr. Toro-Londono and colleagues:

Thanks for submitting your manuscript to PeerJ. I have now received three independent reviews of your work, and as you will see, the reviewers raised some concerns about the research. Nonetheless, these is enough optimism for me to encourage you to revise your work and resubmit. Importantly, please ensure that an English expert thoroughly evaluates your revision. Also, the quality of the scientific writing stands for improvement, so please comply with the suggestions raised by the reviewers to improve the writing and presentation style. In particular, please streamline the long-winded Introduction and get to your main point (which needs to be better articulated) as briefly and clearly as possible.

Regarding scientific content, in your revision please describe your methodologies in a little more detail. Reviewer 1 raises several points regarding this. Sampling should always be justified when only a portion of samples are analyzed. Please also ensure that the figures and figure legends are clear and “stand-alone’. Several reviewers struggled with this. The tables also need better explanation in certain places. Please clearly explain the switch in enterotypes, as this is rather confusing.

Accordingly, I am recommending that you revise your manuscript, taking into account all of the issues raised by the reviewers. I look forward to seeing your revision, and thanks again for submitting your work to PeerJ.

Good luck with your revision,

-joe

Reviewer 1 ·

Basic reporting

There are many awkward phrases, improper grammar and incorrect sentence structure throughout the manuscript. Some examples are: Line 34 – “children population in developing countries . . .”. Line 45 – it is not determinant in gut microbiota . . .”. Line 90-91 – educing the second minimal to non-inflammatory . . .
The authors should closely review the text for readability and proper English usage.

The introduction is long and somewhat disjointed. The authors should review the introduction and look toward streamlining it as well as providing a better focus.

Experimental design

It is unclear why 23 samples were selected for study from the more that 200 samples that were collected. How were the 23 chosen? Did only a small percentage of the samples contain parasites? Specifics concerning the selection process are necessary.

How were the controls defined? Was it only the lack of detectable parasites?
More information about the patient samples would have been useful. Had any of the individuals been recently treated for a parasitic infection? They could be negative for parasites, but have significant changes due to a previous infection.

It appears that only single samples from individuals were used for the study. Would multiple samples taken during and after clearance/treatment of the parasites provide more robust data?

Validity of the findings

It is not clear why figure 1 was included in the manuscript. It should be removed or its utility clearly demonstrated in the text.

The inclusion of the map in figure 2 would be more useful if there was a discussion of what, if any, significance there was for the location of the child care centers. Were there differences in socio-economic levels, urban versus periurban or suburban location. Otherwise it is unclear what the map adds to the manuscript.

The authors discuss the change in enterotypes, but it is not made entirely clear if there is some significance to this switch. Particularly with regards to the specific individuals in the study.

Reviewer 2 ·

Basic reporting

no comment

Experimental design

no comment

Validity of the findings

no comment

Additional comments

In the present study authors sampled the infected stools of 23 children from 4 different children’s care-centers in Medellin, Colombia and identified the eukaryotic parasites by traditional and molecular methodologies. Subsequently, 16S rDNA sequencing was performed to measure the microbial diversity, and later, influence of the different species of parasites on microbial diversity of gut was calculated. This study finds significant influence of the intestinal parasites on the diversity of gut microbiota. Though, study is encouraging, several sections of manuscript need significant improvement as mentioned below:

1. Dietary habits of individuals have not been reported in the manuscript. Dietary habit contributes to the changes in the composition of gut microbiota, which can have significant role on the findings of this study.
2. First and last two sentences of the abstract need to be re-written.
3. Introduction section is too lengthy and unfocused. Several facts have been unnecessarily discussed. For example: Line 89-197 can be reduced drastically.
4. Last paragraph of the introduction is not systematically written.
5. Discussion section is also lengthy. Especially, most of facts discussed in lines 353-416 can be removed.
6. At several places, including the figure legends, microbiota has been referred as microbiome.
7. Legend of figure 1 should be changed, something like “Schematic of human gut microbiota”.
8. Legend and description of the figure 6 should be written in more detail.
9. Legend of Table 1 can be changed to “Description of samples used in this study”.
10. Legend of tables 2 & 3 is well focused and descriptive.

Reviewer 3 ·

Basic reporting

The structure of manuscript is in according with journal criteria however moderate English changes are needed.
In the introduction, the role of protozoa in diarrhea is better described in the GEMS and MAL-ED studies (Kotloff 2013, Platts-Mills 2015).These studies highlight the role of protozoan as etiologic agents of enteric infections in the first two years of age, particularly Cryptosporidium spp. associated to moderate-to- severe diarrhea and Giardia lamblia in asymptomatic infected infants. Entamoeba h. has a minimal role.

Experimental design

The study is in according with the scope of the journal.
Since helminths and protozoa could shift the intestinal microbiota in different ways I suggest to analyze children exclusively infected by helminths in a separate group instead of mixed infections.

Validity of the findings

In the results data are clear and well presented. The tables and figures are relevant and well described. However I suggest another table with the clinical variables including nutritional status and diarrhea, so the readers can have a better perspective to interpret the results.
The discussion is well formulated and the main points are discussed. Since Giardia and helminths have different effects on GM it should be better addressed in separated paragraphs. The consequences of microbiota type-II on growth and neurodevelopment in children also deserve more discussion.
Limitations of the study should be point out.

Conclusions are clear and in according with the aim of study.

Additional comments

The paper addressed the interaction parasites-gut microbiota in children, which is of great interest since contributes to the understanding of immunomodulatory effects and pathogenesis of protozoa and helminth infection on pediatric population.

---

## Round 0.2 · Minor Revisions

Dear Dr. Toro-Londono and colleagues:

Thanks for re-submitting your manuscript to PeerJ, and for addressing the concerns raised by the reviewers. I have now received three independent reviews of your revision, and as you will see, all are mostly favorable. Well done! Nonetheless, the reviewers raised some relatively minor concerns about the research, and areas where the manuscript can still be improved. I agree with the reviewers, and thus feel that these concerns should be adequately addressed before moving forward.

Importantly, please ensure that an English expert has proofed your manuscript.

Therefore, I am recommending that you revise your manuscript accordingly, taking into account all of the issues raised by the reviewers. I do believe that your manuscript will be ready for publication once these issues are addressed.

Good luck with your revision,

-joe

Reviewer 1 ·

Basic reporting

The English and grammar in this revised manuscript have been improved, but there are still a number of awkward phrases or grammatical issues. For example: Lines 57-60, line 108, lines 154-156, lines 249-251, lines 410-411. Please thoroughly review the manuscript.

The investigators response for justifying the inclusion of Figure 1 does not adequately supports its inclusion. The figure is not needed.

The map of the area is really not needed. The proximity of the the child care centers can be easily described in the text. It was not not clear if all of the centers were in urban or peri-urban locations.

Experimental design

The authors included more information about how the samples were chosen, but there is some confusion. The number of samples that were selected for analysis was 23, but the different groups that were identified by the authors do not add up to 23. Please explain.

Validity of the findings

The authors suggest that populations from the same area will have the same enterotypes. They should provide references to support this contention.

It is probably not accurate to describe the diet in Medellin as westernized based upon the enterotypes. Wouldn't it be more appropriate to survey the diets (or provide references to information about the diets) and then make a statement about the relationship to the enterotypes based upon the diet?

Additional comments

While improved, there are still several issues that the authors need to address.

Reviewer 2 ·

Basic reporting

no comment

Experimental design

no comment

Validity of the findings

no comment

Additional comments

Authors have justified my previously raised issues. Improved manuscript is well suited for the acceptance.

Reviewer 3 ·

Basic reporting

The introduction is still large and all manuscript needs english revision.

Experimental design

From 290 children only 23 had positive results for enteric parasite. This small sample will not allow valid conclusions.
A new design is needed in order to have a better sample of single infections.

Validity of the findings

Validity of findings is limited for the small sample and mixed infections.

Additional comments

This manuscript need major revisions.

---

## Round 0.3 · accepted · Accept

Dear Dr. Toro-Londono and colleagues:

Thanks for re-submitting your manuscript to PeerJ, and for addressing the concerns raised by the reviewers. I now believe that your manuscript is suitable for publication. Congratulations! I look forward to seeing this work in print, and I anticipate it being an important resource for parasitologists studying the relationship between parasites and bacteria colonizing the same host. Thanks again for choosing PeerJ to publish such important work.

-joe

#